# Longitudinal Circulating Tumor DNA Analysis in Blood and Saliva for Prediction of Response to Osimertinib and Disease Progression in EGFR-Mutant Lung Adenocarcinoma

**DOI:** 10.3390/cancers13133342

**Published:** 2021-07-03

**Authors:** Chul Kim, Liqiang Xi, Constance M. Cultraro, Fang Wei, Gregory Jones, Jordan Cheng, Ahmad Shafiei, Trinh Hoc-Tran Pham, Nitin Roper, Elizabeth Akoth, Azam Ghafoor, Vikram Misra, Nina Monkash, Charles Strom, Michael Tu, Wei Liao, David Chia, Clive Morris, Seth M. Steinberg, Hadi Bagheri, David T. W. Wong, Mark Raffeld, Udayan Guha

**Affiliations:** 1Thoracic and GI Malignancies Branch, CCR, NCI, NIH, Bethesda, MD 20892, USA; cultrarc@navmed.nci.nih.gov (C.M.C.); nitin.roper@nih.gov (N.R.); elizabeth.akoth@nih.gov (E.A.); azam.ghafoor@nih.gov (A.G.); vikram.misra@nih.gov (V.M.); nmonkash@nyit.edu (N.M.); 2Laboratory of Pathology, CCR, NCI, NIH, Bethesda, MD 20892, USA; xil2@mail.nih.gov (L.X.); phamtrin@mail.nih.gov (T.H.-T.P.); mraff@mail.nih.gov (M.R.); 3School of Dentistry, University of California, Los Angeles, Los Angeles, CA 90024, USA; fwei@dentistry.ucla.edu (F.W.); jcheng1@g.ucla.edu (J.C.); dtww@ucla.edu (D.T.W.W.); 4Inivata, Cambridge CB21 6GS, UK; greg.jones@inivata.com (G.J.); clive.morris@inivata.com (C.M.); 5Radiology and Imaging Sciences, Clinical Center, NIH, Bethesda, MD 20892, USA; ahmad.shafiei@nih.gov (A.S.); hadi.bagheri@nih.gov (H.B.); 6Liquid Diagnostics LLC, San Clemente, CA 92673, USA; charlesmstrom@gmail.com (C.S.); emtu@ucla.edu (M.T.); 7EZLife Bio Inc., Los Angeles, CA 91324, USA; wei.liao@ezlife.bio; 8Department of Pathology and Laboratory Medicine, UCLA, Los Angeles, CA 90095, USA; dchia@mednet.ucla.edu; 9Biostatistics and Data Management Section, NCI, NIH, Bethesda, MD 20892, USA; steinbes@mail.nih.gov

**Keywords:** ctDNA, EGFR, osimertinib, NSCLC

## Abstract

**Simple Summary:**

ctDNA assay is a promising non-invasive method to detect genomic alterations associated with lung cancer. In this prospective study of 25 patients with EGFR-mutant lung adenocarcinoma receiving osimertinib, ctDNA progression predated radiographic progression by 118 days in 11 of 20 patients with disease progression. Saliva-based ctDNA analysis and plasma NGS detected additional patients with ctDNA progression preceding clinical progression, suggesting the potential complementary roles of different ctDNA detection methodologies. Baseline mutant ctDNA level predicted progression-free survival while tumor volume measurements by volumetric CT did not. Serial ctDNA analysis of plasma and saliva is a clinically useful tool to monitor response and resistance to osimertinib.

**Abstract:**

*Background*: We assessed whether serial ctDNA monitoring of plasma and saliva predicts response and resistance to osimertinib in EGFR-mutant lung adenocarcinoma. Three ctDNA technologies—blood-based droplet-digital PCR (ddPCR), next-generation sequencing (NGS), and saliva-based EFIRM liquid biopsy (eLB)—were employed to investigate their complementary roles. *Methods*: Plasma and saliva samples were collected from patients enrolled in a prospective clinical trial of osimertinib and local ablative therapy upon progression (NCT02759835). Plasma was analyzed by ddPCR and NGS. Saliva was analyzed by eLB. *Results*: A total of 25 patients were included. We analyzed 534 samples by ddPCR (*n* = 25), 256 samples by NGS (*n* = 24) and 371 samples by eLB (*n* = 22). Among 20 patients who progressed, ctDNA progression predated RECIST 1.1 progression by a median of 118 days (range: 61–272 days) in 11 (55%) patients. Of nine patients without ctDNA progression by ddPCR, two patients had an increase in mutant *EGFR* by eLB and two patients were found to have ctDNA progression by NGS. Levels of ctDNA measured by ddPCR and NGS at early time points, but not volumetric tumor burden, were associated with PFS. *EGFR*/*ERBB2*/*MET/KRAS* amplifications, *EGFR* C797S, *PIK3CA* E545K, *PTEN* V9del, and *CTNNB1* S45P were key resistance mechanisms identified by NGS. *Conclusion*: Serial assessment of ctDNA in plasma and saliva predicts response and resistance to osimertinib, with each assay having supplementary roles.

## 1. Introduction

Non-small cell lung cancer (NSCLC) is the most common type of lung cancer, accounting for 85% of lung cancer cases, while lung adenocarcinoma is the most common NSCLC histology. The selection of systemic treatment based on molecular profiling has become an essential component of clinical care in patients with advanced NSCLC [1]. Tumor tissue has served as the main source for molecular profiling. While genotyping tumor tissue could provide a wealth of information on somatic genetic alterations, sampling tumor tissue is often not feasible and may lead to morbidity [2]. Tumor tissue is also subject to intra- and inter-tumor heterogeneity [3,4]. Multi-region and temporal sequencing of tumor tissue during the course of the third-generation epidermal growth factor receptor (EGFR) tyrosine kinase inhibitor (TKI) osimertinib has identified significant heterogeneity in resistance mechanisms [4]. Such depth of analysis required to identify driver mutations and genomic alterations mediating resistance to targeted therapies remains a challenge for patients undergoing treatment.

Circulating tumor DNA (ctDNA) has emerged as a promising non-invasive tool to detect various genomic alterations associated with a broad array of malignancies, including NSCLC [5,6]. The advent of sensitive genomic techniques such as droplet digital polymerase chain reaction (ddPCR) and tagged-amplicon sequencing (TAmSeq) has enabled detection and enumeration of somatic genetic aberrations in blood samples [7,8,9]. A high level of concordance is observed between the mutations detected in plasma DNA and those found in the corresponding primary NSCLC [10]. Detection of sensitizing *EGFR* mutations as well as the *EGFR* T790M mutation by plasma genotyping has been incorporated into routine clinical practice [8].

Emerging evidence suggests that ctDNA can be used to monitor treatment response. In patients with EGFR-mutant NSCLC, decreases in plasma mutant *EGFR* ctDNA were associated with improved treatment outcomes [11,12,13,14]. Serial assessment of plasma *EGFR* mutations demonstrated that dynamic changes in plasma ctDNA levels correlated with the therapeutic efficacy of EGFR-TKI therapy [15,16,17,18]. While most previous studies utilized single-gene testing, next-generation sequencing (NGS)-based multigene panel testing may provide advantages given the ability to detect and track multiple mutations.

Previous studies have mainly used plasma samples for ctDNA detection, but other biofluids such as saliva could provide alternative sources for ctDNA assessment. For example, using an electrochemical liquid biopsy technology called electric field-induced release and measurement (EFIRM), the utility of ctDNA detection in saliva samples has been assessed. The EFIRM platform was able to detect *EGFR* exon 19 deletion and L858 R in both advanced and early-stage NSCLC [19,20]. In addition, EFIRM has been shown to detect ultrashort ctDNA (usctDNA) with the size of 40–60 base pairs [21]. Whether biofluids other than blood can be used for monitoring of treatment response is largely unexplored.

Here, we assessed whether longitudinal mutant ctDNA monitoring using blood and saliva samples could reflect response and resistance to osimertinib, a third-generation EGFR-TKI, and tumor burden measured by three-dimensional volumetric computed tomography (CT) in the setting of a prospective clinical trial of local ablative therapy (LAT) for oligoprogressive, and EGFR-mutant NSCLC upon treatment with osimertinib. Three ctDNA platforms—blood-based droplet-digital PCR (ddPCR) and next-generation sequencing (NGS) and saliva-based EFIRM liquid biopsy (eLB)—were used to investigate their complementary roles.

## 2. Materials and Methods

### 2.1. Study Population and Specimen Collection

In a clinical trial for patients with EGFR-mutant lung adenocarcinoma at the National Cancer Institute/National Institutes of Health (NCT02759835), we are investigating the safety and efficacy of osimertinib re-initiation after local ablative therapy (LAT) following oligoprogression (≤5 sites of progression) while on osimertinib treatment. We also aim to examine mechanisms of osimertinib resistance using multiomics analyses of tumor and liquid biopsies. After the principal investigator left the institution, new patient enrollment was halted.

Patients with no prior EGFR-TKI therapy (cohort 1) or those with T790M-positive NSCLC after first-line EGFR-TKI treatment (cohort 2) started osimertinib. Upon oligoprogression, patients underwent LAT (surgery, radiation therapy, radiofrequency ablation), after which osimertinib treatment was resumed. Patients previously treated with osimertinib outside of the NIH Clinical Center eligible for LAT upon the development of oligoprogression were also enrolled (cohort 3). The schema of the clinical trial is shown in Figure 1. Liquid biopsies (blood, saliva, and urine) were procured at every clinic visit in this clinical study, in addition to the collection of samples on day 7 during cycle 1. Radiographic tumor assessment was performed every 2 cycles. A cycle of osimertinib treatment was 21 days for initial 8 patients, and subsequently, the protocol was amended to be every 28 days.

Blood was collected in EDTA-containing tubes and stored on ice until processing. Plasma was separated by centrifugation at 4 °C for 10 min at 2000× *g* and the plasma cleared by centrifugation at 4 °C for 10 min at 14,000× *g*. Saliva was collected in Pure•SAL^TM^ (Oasis Diagnostics, Vancouver, WA, USA) and stored at −80 °C without further processing.

The study was conducted in accordance with the Declaration of Helsinki.

### 2.2. Tumor Volumetric Measurements

All target lesions for each patient were identified at each tumor assessment using computed tomography (CT). Then, to estimate the calculated tumor burden in the body, we segmented and measured the volumes of all soft tissue lesions with the following size inclusion limits. For lung lesions, a long axis of ≥7 mm was included. For all other soft tissue lesions, a long axis of ≥10 mm, and for lymph nodes, a short axis of ≥10 mm was considered. All of these lesions were manually segmented, and the volume was measured using Carestream PACS (Vue PACS version 12.1, Carestream Health, Rochester, New York, NY, USA; Appendix A). These lesion volumes were captured even if the lesion got smaller in the follow-up scans.

### 2.3. ddPCR

Cell-free DNA (cfDNA) was isolated from 2–4 mL plasma samples using the MagMax cell-free DNA isolation kit with the KingFisher Prime Duo instrument (ThermoFisher) according to the manufacturer’s instructions. Circulating tumor DNA (ctDNA) detection was performed on a BIO-RAD QX200 ddPCR system using the custom PrimePCR ddPCR mutation detection assay (BIO-RAD, Hercules, CA, USA) for specific *EGFR* mutations originally identified in a patient’s tumor specimen (Appendix A). Each PCR reaction contained 10 µL of 2× ddPCR supermix for probes (no dUTP), 1 µL 20× mutant primers/probe mix (FAM) and wild type primers/probe mix (HEX) mix, 1 µL nuclease-free water, and 8 µL of cfDNA. The assay was performed in duplicate. The presence of mutant DNA copies and the fractional abundance of the mutant allele were determined with QuantaSoft v.1.7 (BIO-RAD). Mutant *EGFR* copy number was normalized to plasma volume and is expressed as copies/mL plasma. ctDNA progression by ddPCR was defined using the following criteria: (1) conversion from negative to positive AFs, or (2) increase in AFs two consecutive timepoints by more than 10%.

### 2.4. NGS

For the next generation sequencing testing, the results were generated using an amplicon-based sequencing platform, InvisionFirst™-Lung, which analyzes somatic changes within 36 cancer genes (Appendix A). Testing was performed as previously described [22,23]. Briefly, sequencing libraries were created from extracted cfDNA using a two-step amplification process and were sequenced on the Illumina NextSeq 500 platform. Using a proprietary analytical pipeline, genomic alterations were identified and reported. Copy number was determined by the extra signal seen in a gene compared to the signal in other genes. The imbalance was calculated by a formula where the amplicon depths of the PCR reactions for a given gene were normalized across the amplicon depths for the rest of the genes in that sample and across the other samples in the run. This method has been previously validated [23].

### 2.5. EFIRM

ctDNA in salivary samples were analyzed using EFIRM liquid biopsy (eLB) assay (EZLife Bio, Woodland Hills, CA) [19]. *EGFR* mutations (Exon 19 deletion, L858R, and T790M) were assessed using paired capture and detector probes (Integrated DNA Technologies, San Diego, CA, USA) (Appendix A). Capture probes were designed as 13–14 base pair (bp) single-stranded DNA oligomers that hybridize with the mutation sequence of ctDNA fragments. The capture probe and detector probe sequences are contiguous and provide a two-factor specificity requirement with the target molecule so that both probes must hybridize to the target before a signal can be generated. There is a 71 bp non-specific poly-A tail at the 5′ end of the capture probe to create distance from the polymer to encourage binding with the target from the solution. The sequence of the probes was optimized by maximizing sensitivity (signal to background ratio for lower concentrations) and specificity (signal with a wildtype target) using empirical data from oligomer DNA targets (Integrated DNA Technologies, San Diego, CA, USA).

### 2.6. Statistical Considerations

Descriptive statistics were used to describe patient characteristics. Correlations between ddPCR, NGS, eLB, and tumor volume were examined using Spearman rank-order correlation. In addition, Spearman correlation was used to determine the correlation between change in volume and change in AF and copy number from baseline to day 7, 21/28, or day 42/56. Correlations such that |r| > 0.70 would be considered strong; if 0.50 < |r| < 0.70, the correlation was considered moderately strong, if 0.30 < |r| < 0.50, the correlation was weak to moderately-strong, and if |r| < 0.30, the correlation was weak. The Kaplan–Meier method was used to assess the probability of progression-free survival (PFS) as a function of time. The patients were divided into two categories of approximately equal size to assess the association between the category and PFS. Categories that were naturally occurring as zero vs. any positive value remained that way. A log-rank test was used to determine the statistical significance of the difference between the two groups. No formal adjustment for multiple comparisons was performed. Instead, the actual *p*-values were interpreted as the strength of the evidence that the two groups have differing PFS, without formally declaring an arbitrary threshold for statistical significance. The hazard ratios for comparing two groups were determined from a Cox proportional hazards model. Statistical analyses were performed using Prism version 9.0.1 (GraphPad Software, San Diego, CA, USA) and SAS version 9.4 (SAS Institute, Cary NC, USA).

## 3. Results

### 3.1. Patients and Clinical Samples

In this study, we focused on 25 patients with sufficient clinical follow-up and biospecimens. Fifteen (60%) patients (cohort 1) received osimertinib as first-line treatment (cohort 1) (Appendix A). Ten (40%) patients (7 in cohort 2 and 3 in cohort 3) were treated with osimertinib for T790M-positive NSCLC. A total of 523 blood samples from 25 patients were analyzed by ddPCR. Tagged-Amplicon Sequencing NGS assay was performed on a select 256 blood samples from 24 patients. We analyzed 371 saliva samples from 22 patients by eLB.

### 3.2. Volumetric Tumor Measurements

A total of 265 CT scans for 25 patients at baseline and follow-up visits were evaluated. Calculated tumor volumes of 112 lesions (the range of the number of lesions per patient: 1–26, median 3) at multiple time points (the range of the number of time points per patient: 4–19, median 12) were measured (Appendix A). A total of 1094 tumor lesion volumes were captured, including 414 in lung, 312 in liver, 189 in lymph nodes, and 179 in other parts of the body.

### 3.3. Baseline Detection of Plasma ctDNA and Association between Baseline Mutant EGFR Plasma ctDNA Levels and Tumor Burden

EGFR mutations (sensitizing and/or T790M) were detected by ddPCR in 21 (88%) of 25 patients at baseline. Among the four patients without detectable EGFR mutations, one patient (LAT009) was a patient in Cohort 3, already on treatment with osimertinib, who had disease mainly in the brain, including disease progression in the brain which was treated with LAT (surgery and radiation therapy as per the protocol). Two patients (LAT019, LAT025) had a low disease burden with a calculated tumor volume of 1.86 cm^3^ and 7.8 cm^3^, respectively. One patient (LAT026) in Cohort 2 with a high calculated tumor burden of 148 cm^3^ did not have detectable EGFR mutations. In patients with detectable EGFR mutations in the baseline plasma sample (*n* = 21), the mean quantity of cfDNA was 236.0 (range: 9.8–2040.0). In those without detectable EGFR mutations at baseline (*n* = 4), the mean quantity of cfDNA was lower at 30.9 (range: 20.9–41.1).

The AFs of sensitizing EGFR mutations measured by ddPCR and NGS at baseline were weakly to moderately well-correlated with baseline tumor volume (Figure 2A,B; Spearman ρ = 0.36 with *p* = 0.074 for ddPCR and Spearman ρ = 0.45 with *p* = 0.026 for NGS). The Spearman correlation coefficient between sensitizing EGFR mutation copy number and baseline tumor volume was ρ = 0.45 with *p* = 0.024.

### 3.4. Correlations between ddPCR; NGS; eLB, and Calculated Tumor Volume

After compiling ddPCR AF, NGS AF, and eLB current signal for EGFR mutations at each time point, we examined the correlations between ctDNA detection modalities for EGFR mutation detection and between each ctDNA detection modality and calculated tumor volume. A strong correlation between ddPCR- and NGS-detected mutant EGFR AFs was found (Appendix A and Figure 3A; Spearman ρ = 0.93; *p* < 0.001). Plasma ddPCR and NGS were weakly correlated with the saliva eLB mutant EGFR detection assay (Spearman ρ = 0.24 with *p* < 0.001 and ρ = 0.24 with *p* = 0.003, respectively; Figure 3B,C). Each ctDNA detection modality was weakly or weakly to moderately correlated with calculated tumor volume (Spearman ρ = 0.35, 0.46, and 0.28 for ddPCR, NGS, and eLB, respectively; *p* < 0.001 for all).

### 3.5. Dynamic ctDNA Changes Reflect Treatment Response and Emergence of Resistance

We investigated whether serial monitoring of ctDNA could predict osimertinib treatment response and the emergence of resistance (Figure 4). Among 20 patients who had RECIST 1.1 progression, ctDNA progression predated RECIST 1.1 progression by a median of 118 days (range: 61–272 days) in 11 (55%) patients. NGS and eLB also showed similar patterns of ctDNA rise before radiographic progression. Of the nine patients without ctDNA progression by ddPCR, two patients had an increase in EGFR mutation-level by eLB (LAT006, LAT026) and two patients were found to have ctDNA progression by NGS (increase in PTEN Y88* AF in LAT007 and increase in TP53 V157F in LAT016). In 5 patients, ctDNA progression did not precede RECIST 1.1 progression (LAT011, LAT020, LAT021, LAT022, LAT025; Appendix A); ctDNA progression lagged radiographic RECIST 1.1 progression in two patients (12 days for LAT011, 10 days for LAT021). In one patient who passed away, likely from a myocardial infarction, they did not have ctDNA nor RECIST 1.1 progression (LAT020). In all patients without radiographic progression (*n* = 5) at the time of ctDNA analysis, there was no evidence of ctDNA rise (Appendix A).

For 22 patients in cohort 1 (*n* = 15) and 2 (*n* = 7), we evaluated whether mutant EGFR AF and copy numbers at baseline, day 7, day 21/28, and day 42/56 measured by ddPCR were associated with PFS on osimertinib (patients in cohort 3 were excluded from this analysis because they presented with osimertinib-resistant disease). For one patient, a day 42/56 value was not available, so the next available value (day 70) was substituted for this value. Mutant EGFR copy numbers at baseline and day 21/28 were predictors of PFS with hazard ratios (HRs) of 2.41 (95% CI: 0.98–5.91, *p* = 0.048) and 3.39 (95% CI: 1.21–9.50, *p* = 0.014), respectively (Figure 5A,B). Median PFS was 19.5 months in patients with low mutant EGFR copy number (95% CI: 11.2–31.2 months) vs. 8.9 months in those with high mutant EGFR copy number (95% CI: 3.6–14.7 months) at baseline. On day 21, median PFS was 17.4 months in patients with low mutant EGFR copy number (95% CI: 7.4–31.2 months), while it was 10.1 months in those with high mutant EGFR copy number (95% CI: 3.4–15.5 months). Next, we used NGS to interrogate the association between mutant EGFR AF at baseline and day 7 with PFS. Day 21/28 and day 42/56 were not assessed because NGS was not performed in most patients at these time points. Baseline mutant EGFR AF was not associated with PFS, but mutant EGFR AF on day 7 was associated with PFS (Figure 5C,D). For mutant EGFR AF on day 7, median PFS was 13.9 months in patients with low EGFR AF (95% CI: 6.9–42.4 months), while it was 8.9 months in those with high EGFR AF (95% CI: 3.4–14.7 months). We further interrogated whether tumor volume assessed by volumetric CT measurement, a surrogate of calculated tumor burden, was associated with PFS. Tumor volume at baseline (Figure 5E), day 42/56, the first follow-up scan on treatment, and the difference in tumor burden between baseline and day 42/56 (Figure 5F) were not associated with PFS.

### 3.6. NGS ctDNA Assay (InVisionFirst-Lung) Detects a Broad Array of Genomic Modifications Which May Be Implicated in Osimertinib Resistance

The advantage of NGS over ddPCR for ctDNA detection is the ability to identify co-occurring mutations and copy number changes. Co-occurring genomic alterations in TP53 (*n* = 13), CDKN2A (*n* = 2 including a germline CDKN2A mutation in LAT011), PIK3CA (*n* = 1), PTEN (*n* = 1), and amplifications in EGFR (*n* = 3), ERBB2 (*n* = 1), KRAS (*n* = 1) were detected by NGS in baseline plasma samples from 21 patients without prior exposure to osimertinib (Figure 6). In all patients with detectable ctDNA via NGS (*n* = 23), amplifications of EGFR (*n* = 7), ERBB2 (*n* = 4), MET (*n* = 4), and KRAS (*n* = 2), and somatic mutations, including PIK3CA/PTEN mutations (*n* = 5), EGFR C797S mutation (*n* = 1), and CTNNB1 mutation (*n* = 1) were identified as key resistance mechanisms upon osimertinib treatment. A patient (LAT025) who underwent lung resection for oligoprogressive disease was found to have new KRAS G12C and TP53 227–228:SD/X after the surgery without the original EGFR exon 19 deletion mutation, suggesting that the resected tumor likely represent a new primary tumor or a concomitant second primary tumor that grew during the course of treatment. Details of genomic alterations and allele frequencies from each patient at various time points are available as Appendix A.

## 4. Discussion

Osimertinib is becoming the standard treatment option for EGFR-mutant NSCLC. While its therapeutic advantages over earlier-generation EGFR-TKIs have been demonstrated in a randomized trial (FLAURA trial) [24], the emergence of resistance to osimertinib is inevitable in virtually all patients. Current decision-making for osimertinib response mainly relies on the radiographic evaluation of tumors which does not inform the dynamic biological processes of tumor evolution. Tumor biopsies obtained at progression are used to identify genomic alterations and define resistance mechanisms, but they are invasive, not always feasible, and associated with inherent challenges imposed by tumor heterogeneity.

In this prospective study, we demonstrated that longitudinal assessment of ctDNA can be used to predict clinical outcomes in patients with EGFR-mutant lung adenocarcinoma treated with osimertinib. Mutant *EGFR* copy number from ctDNA in plasma obtained at baseline and day 21 after initiating osimertinib treatment was strongly associated with PFS, with higher copy number portending worse PFS. Similarly, in clinical trials of osimertinib (AURA3, FLAURA, TATTON), clearance of mutant *EGFR* in plasma at weeks 3 and 6 was associated with better PFS [25,26,27] (presented at ASCO annual meeting 2018 and 2019, and AACR 2020, respectively). In the FLAURA trial, the lack of detection of mutant *EGFR* from ctDNA by cobas plasma testing at baseline was associated with prolonged PFS [28]. It was suggested that the improved PFS could be due to cobas plasma-negative patients having a low tumor burden. Contrary to this suggestion, baseline tumor burden analyzed by volumetric CT was not associated with PFS in our study, suggesting tumor burden may not explain the better prognosis in those with low mutant *EGFR* level at baseline. Lastly, some studies report that tumor volume decrease after EGFR-TKI therapy is associated with survival [29,30], but this was not observed in our study. These findings need validation in future studies that ideally utilize the prospective collection of ctDNA and volumetric assessment of tumor burden.

Importantly, increases in ctDNA with *EGFR* mutation detected by by ddPCR preceded RECIST 1.1 progression by 118 days in 11 of 20 patients with progressive disease. In one patient (LAT013) who had a partial response that lasted less than 3 months, ctDNA progression was noted even prior to partial response per RECIST 1.1, suggesting that ctDNA assessment could serve as a sensitive tool to predict the quality of treatment response. Of the 9 patients without ctDNA progression by ddPCR, eLB identified two patients with increasing *EGFR* mutation levels and blood-based NGS revealed molecular progression in two additional patients. Overall, these results highlight the utility of longitudinal ctDNA measurements of patients while on osimertinib treatment in predicting PFS and resistance to EGFR-TKI therapy and the potential complementary roles of different platforms in monitoring disease progression.

Results of ddPCR can be expressed as either absolute values (mutant copies/mL of plasma) or relative values (proportion of mutated copies). We found that absolute quantification of mutant copies was more strongly associated with PFS when compared with AF. Both absolute mutant *EGFR* copy numbers and AFs have been used in the studies of ctDNA in EGFR-mutant NSCLC, and further studies are needed to elucidate which quantification method is better correlated with treatment outcomes.

The results of our study provide insights into ctDNA biology. One of the intriguing findings was that there were patients who had a discrepancy between mutant *EGFR* ctDNA levels and tumor volume. LAT011 and LAT015 both had a high calculated tumor burden but very low mutant *EGFR* AFs detected by either ddPCR or NGS. The first progression in LAT011 did not result in increased mutant *EGFR* AFs. LAT015 had a sharp decrease in tumor volume due to surgery during LAT, but this did not result in a decreased mutant *EGFR* AF. LAT005, a patient in the third cohort, who underwent unilateral adrenalectomy as a method of LAT, had a decreased tumor burden due to the surgery; however, this was not mirrored by a decrease in mutant *EGFR* AF. Instead, the mutant *EGFR* AF continued to increase even after LAT, suggesting underlying systemic progression.

It should also be noted that a fraction of patients did not experience ctDNA progression prior to RECIST 1.1 progression. These cases highlight that not all tumors “shed” ctDNA into the bloodstream. At the other extreme, there are patients such as LAT001, who had relatively low tumor burden at baseline, and even at first progression but had a quite high mutant *EGFR* AF that reduced to zero by day 42. This is an example of a tumor that releases high levels of mutant ctDNA into the circulation. In our cohort of patients, high mutant *EGFR* ctDNA copy number at baseline and increased clearance of ctDNA, as evidenced by a lower mutant *EGFR* AF as early as Day 7 of treatment, but not the tumor volume, were associated with PFS, suggesting that ctDNA “shedding” may indicate the existence of more aggressive tumors that are likely to develop resistance to therapy faster. The mechanisms by which cancer cells release ctDNA are not fully elucidated [31], and more research is needed to better understand the nature and origin of ctDNA.

We found a very strong correlation between ddPCR and NGS for mutant *EGFR* ctDNA quantitation. While ddPCR is a fast and cost-effective method to detect and quantify cancer-specific mutant ctDNA, NGS provides several advantages over ddPCR for ctDNA analysis. Importantly, NGS enables the simultaneous detection of a broader range of genomic alterations (point mutations, indels, copy number variations, and gene rearrangements) without prior knowledge of the target [32]. We noted co-occurring oncogenic events including *TP53* mutations, amplifications of *EGFR*, *ERBB2*, and *KRAS*, *PIK3CA/PTEN* alterations, and *CDKN2A* alterations, which is consistent with previous work [33]. *EGFR*/*ERBB2*/*MET/KRAS* amplifications, *EGFR* C797S, *PIK3CA* E545K, *PTEN* V9del, and *CTNNB1* S45P were key resistance mechanisms identified by NGS. In 7 patients, there were at least two putative resistance mechanisms, suggesting combination therapies may be necessary to overcome osimertinib resistance. Similar conclusions were derived from multi-region and temporal sequencing of tumors developing resistance to osimertinib in the context of this clinical study [4]. In one patient (LAT016), *EGFR* V726M appeared during treatment with osimertinib, which has been suggested as a potential mechanism of resistance [34]. However, it was only detected on day 280, well before RECIST progression (day 783), and therefore, the significance of the mutation is uncertain. One patient was found to have *CTNNB1* S45P at the time of proregression. The Wnt/β-catenin signaling pathway has been implicated in mediating resistance to EGFR-TKI therapy [35,36]. Whether combined EGFR and β-catenin inhibition could overcome resistance in patients with alterations in the Wnt/β-catenin signaling pathway warrants further investigation.

To the best of our knowledge, this is the first study to serially monitor mutant *EGFR* ctDNA in saliva samples and correlate the level of mutant *EGFR* in saliva to treatment response and resistance. Saliva offers a non-invasive source of ctDNA, making frequent sampling feasible. The saliva collection kit used here (Pure•SAL^TM^) can be used by patients at home and samples shipped to the laboratory for ctDNA detection. The ctDNA dynamics during osimertinib therapy detected by eLB analysis of saliva, in most cases, resembled those of ddPCR and NGS. It is notable that in two patients, ctDNA progression was not identified by plasma ddPCR or NGS, but eLB detected increasing levels of *EGFR* mutations in saliva, which suggests the complementary roles of different ctDNA methodologies in predicting treatment response and resistance to EGFR-TKI therapy. Limitations of the assay, including its semi-quantitative nature, however, should be noted. Further validation of saliva as a ctDNA source and the utility of its analysis by eLB technology is warranted.

Our study has some limitations. First, although 534, 256, and 371 samples were analyzed by ddPCR, NGS, and eLB assays, respectively, the number of patients included in this study is relatively small. Second, we conducted the study to gain insight into the dynamics of ctDNA using different platforms. However, the nature of the study was exploratory and descriptive without formal hypothesis testing. Third, while we show that longitudinal ctDNA measurements can be a clinically useful tool to predict clinical response and disease progression, large prospective studies, particularly randomized trials, are required to determine whether therapeutic modification based on ctDNA analysis provides clinical benefit. Also, to determine the ideal platforms and the frequency of ctDNA testing, further prospective research is required. One technology and one biofluid (e.g., blood) may not be enough to comprehensively track tumor evolution. Combination ctDNA testing methodologies utilizing diverse biofluids may have benefits in terms of comprehensiveness, but logistical considerations and costs of tests should be considered. Fourth, we acknowledge that no standardized definitions of ctDNA progression exist. The definition of ctDNA progression used in our study has not been validated. Lastly, the finding of a lack of correlation of tumor volume measured by volumetric CT with PFS needs to be validated in larger cohorts of patients with different genotypes. This merits further studies for correlation of baseline mutant driver oncogene ctDNA copy number/AF and tumor burden measured by volumetric CT with PFS and overall survival (OS). Currently, clinical trials are largely based on RECIST 1.1 defined tumor measurements. In light of these findings, novel trial designs are warranted that consider ctDNA copy number or AF as metrics of tumor response and PFS/OS.

## 5. Conclusions

Serial measurements of plasma and saliva ctDNA can be useful for monitoring the treatment response to osimertinib, predict PFS, and for early detection of resistance in patients with EGFR-mutant NSCLC. Whether treatment modification based on ctDNA analysis leads to improvement in survival should be determined in large prospective studies.

## Figures and Tables

**Figure 1 cancers-13-03342-f001:**
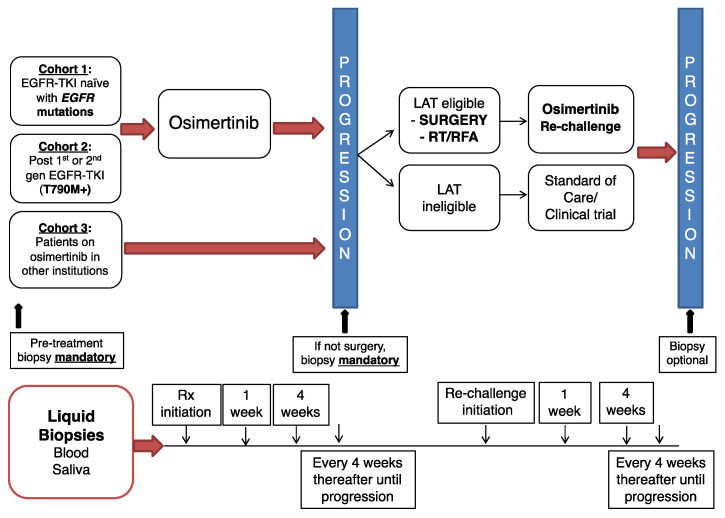
Clinical protocol schema.

**Figure 2 cancers-13-03342-f002:**
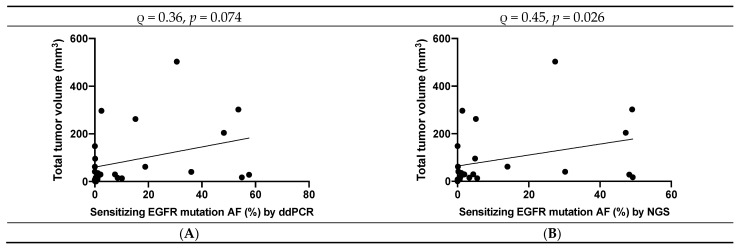
(**A**,**B**) Spearman correlation between *EGFR* mutation AF by ddPCR and tumor volume (**A**). Spearman correlation between *EGFR* mutation AF by NGS and tumor volume (**B**).

**Figure 3 cancers-13-03342-f003:**
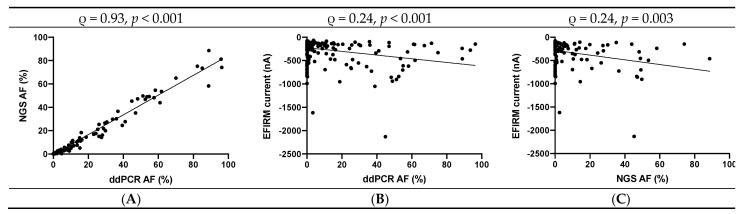
(**A–C**) Correlations between ddPCR, NGS, and EFIRM liquid biopsy (eLB) assays.

**Figure 4 cancers-13-03342-f004:**
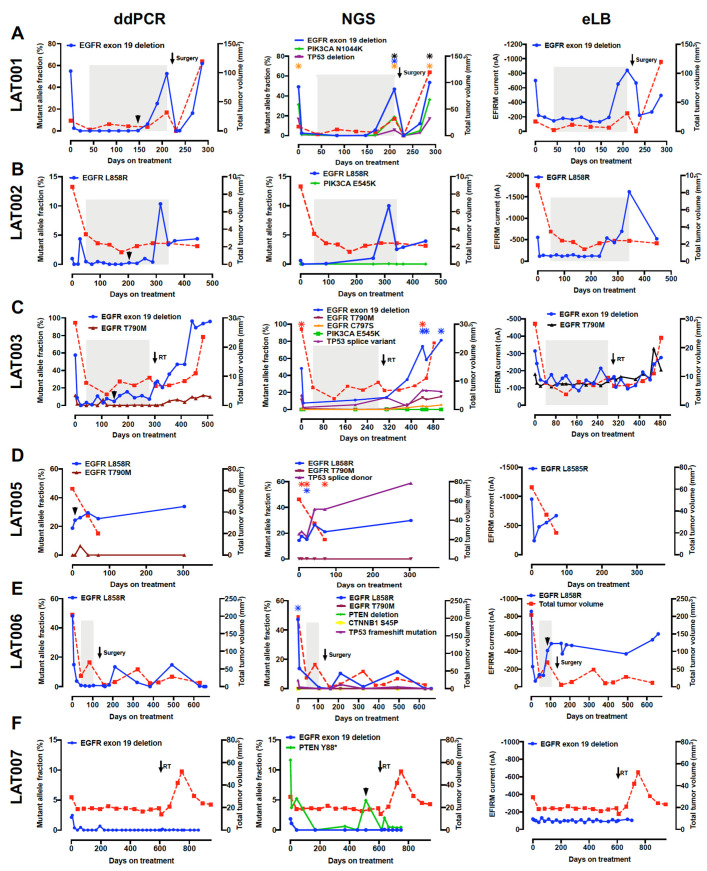
(**A–O**) Serial monitoring of ctDNA by ddPCR, NGS, and eLB for prediction of response and the detection of emergence of resistance. Solid lines represent genomic alterations and red dotted lines represent calculated tumor volume measured by volumetric CT measurement. AFs for ddPCR and NGS and current values for eLB are plotted on the left *y*-axis. Calculated tumor volume measured by volumetric CT measurement is plotted on the right T-axis. Treatment duration is plotted on the x-axis. The shaded gray areas indicate the duration of RECIST 1.1 response (complete or partial response). ctDNA progression, defined as increases in mutant EGFR AF by ddPCR, preceded RECIST 1.1 progression by a median of 118 days (range: 61–272 days) in 11 patients (LAT001, 002, 003, 005, 010, 013, 014, 015, 017, 023, 028). Of the 10 patients without ctDNA progression by ddPCR, 2 patients had an increase in EGFR mutation-level by eLB (LAT006, LAT026), 1 patient had an increase in the AF of PTEN Y88* by NGS (LAT007), and another patient had an increase in TP53 V157F by NGS (LAT016). The arrowhead (in black) indicates the beginning of ctDNA progression.

**Figure 5 cancers-13-03342-f005:**
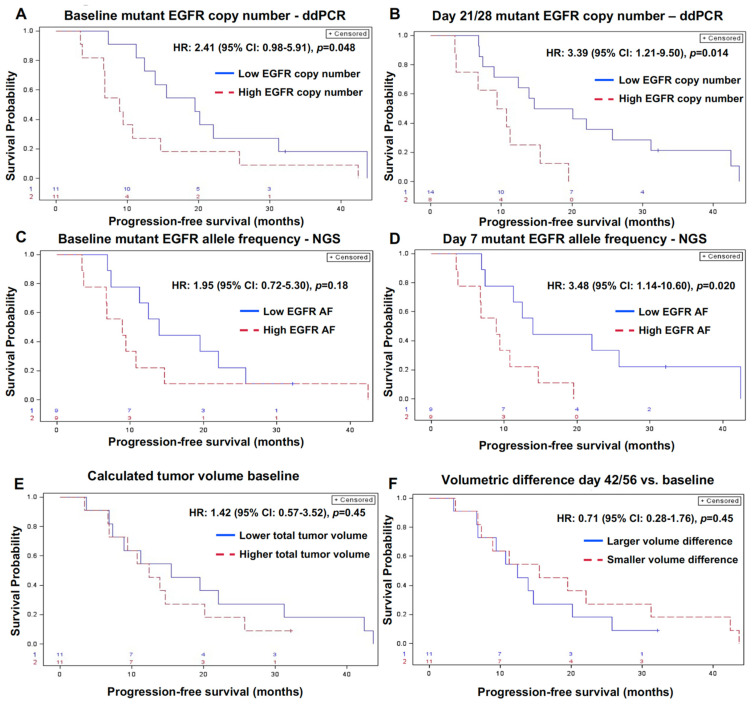
(**A–E**) Progression free survival (PFS) based on the baseline mutant EGFR ctDNA copy number, its clearance on therapy at early time periods on treatment, and baseline tumor volume assessed by volumetric CT. PFS based on mutant EGFR copy number at baseline and on day 21/28 by ddPCR (**A**,**B**), EGFR AF at baseline and day 7 by NGS (**C**,**D**), calculated tumor volume at baseline (**E**), and difference in tumor volume between day 42/46 vs. baseline (**F**).

**Figure 6 cancers-13-03342-f006:**
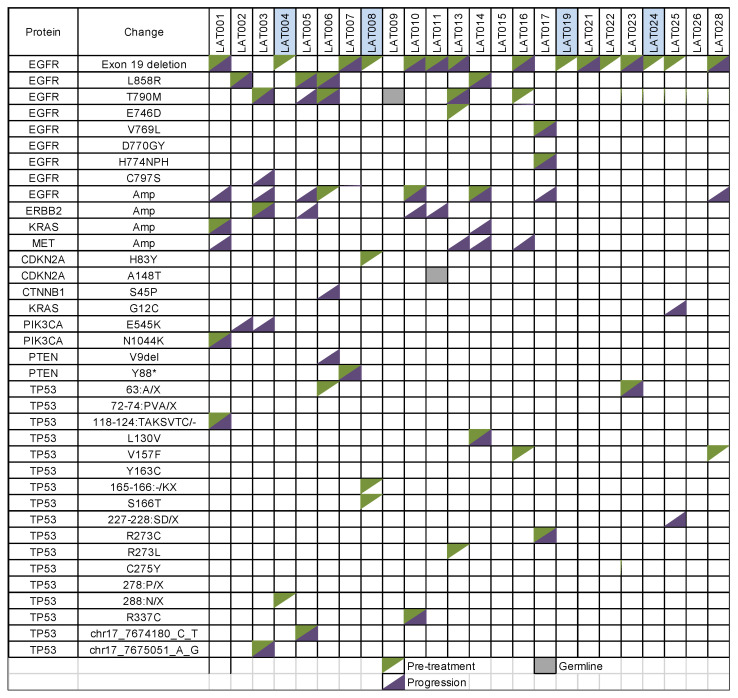
Detection of co-occurring genomic alterations and resistance mechanisms by NGS. Each row represents genomic alterations and columns indicate each patient. Co-occurring mutations at baseline prior to exposure to osimertinib are color-coded in green and genomic alterations seen at progression are in dark purple. Germline mutations are shown in gray. Patients whose disease did not progress at the time of analysis are color-coded in blue (LAT004, LAT008, LAT019, LAT024).

## Data Availability

Not applicable.

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
