# Peer review of "Longitudinal Circulating Tumor DNA Analysis in Blood and Saliva for Prediction of Response to Osimertinib and Disease Progression in EGFR-Mutant Lung Adenocarcinoma"

_cancers, 2021, doi:10.3390/cancers13133342_

Round 1

Reviewer 1 Report

Thank you for inviting me to review this very interesting paper by Dr. Chul Kim and colleagues. First, I would like to commend the authors for the efforts it took for this prospective study with longitudinal assessment of ctDNA to predict the outcomes of patients with EGFR mutant NSCLC. Although the actual number of patients are small, careful assessments of each provide valuable insight as to what could be happening to them at the time of progression. Description of new technology such as eLB also adds value to this paper. Overall, the paper is well written and is scientifically sound. I believe that the clarification on the following points will further strengthen the paper. 

1) What was the hypothesis of the original study (LAT for oligo-progression) and what was it powered to detect?

2) The current paper on ctDNA analysis appears to be part of the original study but WITHOUT a formal hypothesis or statistical considerations with power calculations and appears to be more consistent with a descriptive study which is fine- but should be mentioned in the limitation part of the paper. 

3) If 2 is the case, perhaps the title "Longitudinal circulating tumor DNA analysis in blood and saliva predicts response to osimertinib and disease progression in EGFR-mutant non-small-cell lung cancer" may be misleading. A more conservative title may be something like:  "Longitudinal circulating tumor DNA analysis in blood and saliva to predict response to osimertinib and disease progression in EGFR-mutant non-small-cell lung cancer" 

3) Materials and Methods. 2.3. ddPCR. Please describe further the definition of ctDNA progression. What do you mean by visual inspection? By whom? How much of an increase would be needed to be captured? Is this a standardized technique?

4) Discussion. What are some barriers to implementing this as standard of care? What are the turn around times? Can testing be managed and utilized in real time? 

Reviewer 2 Report

Kim and colleagues adopted digital PCR, NGS and EFIRM to investigate whether serial cfDNA monitoring of plasma and saliva predicts response and resistance to osimertinib in EGFR-mutant lung adenocarcinoma. Their study is comprehensive and could be a good reference to help further development of liquid biopsy in clinical practice. Before publishing, several issues could be further clarified and addressed in the manuscript to make it more clear and complete.

  1. The tumor volumetric measurement only account nodules with a long axis exceed a certain degree of cutoff value. Does include all nodules under the selection criteria improve the correlation between mutant allelic frequency or percentage in cfDNA with tumor volume, response and PFS?

  1. The DNA input used for ddPCR measurement seems to use a specific volume. What is the absolute quantity of cfDNA in these samples and is there any difference between ddPCR positive and negative samples in the baseline data collection?

  1. Since cfDNA is not only from tumor cell but also from the normal cell of human body, how is the reliability of NGS in the determination of gene amplification using cfDNA?

  1. Could authors suggest their preferred scheme in the frequency of sample collection and detection methodology to better help patients monitor disease progression and treatment response?

Reviewer 3 Report

1. The article deals with adenocarcinoma, so I would recommend that the authors change the title of the article from non-small cell lung cancer to adenocarcinoma. 2. Please specify, is osimertinib the only drug that the participants of the experiment took? Usually, a two-component chemotherapy regimen is prescribed, as well as immunotherapy drugs. 3. It is not entirely clear why saliva was used, if the results in comparison with plasma are approximately the same. 
